# BACK-to-MOVE: Machine learning and computer vision model automating clinical classification of non-specific low back pain for personalised management

Thomas Hartley[1], Yulia Hicks[1], Jennifer L. Davies[2,3], Dario Cazzola[4,5], Liba Sheeran[2,3]*

1 School of Engineering, Cardiff University, Cardiff, United Kingdom, 2 School of Healthcare Sciences, Cardiff University, Cardiff, United Kingdom, 3 Biomechanics and Bioengineering Research Centre Versus Arthritis, Cardiff University, Cardiff, United Kingdom, 4 Department for Health, University of Bath, Bath, United Kingdom, 5 Centre for Health, Injury and Illness Prevention in Sport, University of Bath, Bath, United Kingdom

☯ These authors contributed equally to this work.

* sheeranL@cardiff.ac.uk

**Data Availability Statement:** Due to ethical considerations regarding privacy, the video data utilized in this study cannot be shared publicly as

## Abstract

### Background

Low back pain (LBP) is a major global disability contributor with profound health and socio-economic implications. The predominant form is non-specific LBP (NSLBP), lacking treatable pathology. Active physical interventions tailored to individual needs and capabilities are crucial for its management. However, the intricate nature of NSLBP and complexity of clinical classification systems necessitating extensive clinical training, hinder customised treatment access. Recent advancements in machine learning and computer vision demonstrate promise in characterising NSLBP altered movement patters through wearable sensors and optical motion capture. This study aimed to develop and evaluate a machine learning model (i.e., 'BACK-to-MOVE') for NSLBP classification trained with expert clinical classification, spinal motion data from a standard video alongside patient-reported outcome measures (PROMs).

### Methods

Synchronised video and three-dimensional (3D) motion data was collected during forward spinal flexion from 83 NSLBP patients. Two physiotherapists independently classified them as motor control impairment (MCI) or movement impairment (MI), with conflicts resolved by a third expert. The Convolutional Neural Networks (CNNs) architecture, HigherHRNet, was chosen for effective pose estimation from video data. The model was validated against 3D motion data (subset of 62) and trained on the freely available MS-COCO dataset for feature extraction. The Back-to-Move classifier underwent fine-tuning through feed-forward neural networks using labelled examples from the training dataset. Evaluation utilised 5-fold cross-validation to assess accuracy, specificity, sensitivity, and F1 measure.

they may contain potentially identifying and sensitive information, such as tattoos, birthmarks, sex, ethnicity, and unique movement characteristics. Complete de-identification, noise addition, or blocking of portions of the dataset while preserving its usefulness to replicate study results is not feasible. However, external requests for access to the data can be directed to Professor Valerie Sparkes at arthritiscentre@cardiff.ac.uk.

**Funding:** This research was funded by the Wellcome Trust [grant number AC1550IF02]. Funding was awarded to the whole team: LS (PI), TH,YH,DC,JD (Co-Is). Funder website: https://wellcome.org/grant-funding/funded-people-and-projects/institutional-strategic-support-fund. The funder had no role in study design, data collection and analysis, decision to publish, or preparation of the manuscript. For the purpose of Open Access, the author has applied a CC BY public copyright licence to any Author Accepted Manuscript version arising from this submission. This research was part funded by Biomechanics and Bioengineering Research Centre Versus Arthritis (grant number CA20781).

**Competing interests:** The authors have declared that no competing interests exist.

## Results

Pose estimation's Mean Square Error of 0.35 degrees against 3D motion data demonstrated strong criterion validity. Back-to-Move proficiently differentiated MI and MCI classes, yielding 93.98% accuracy, 96.49% sensitivity (MI detection), 88.46% specificity (MCI detection), and an F1 measure of .957. Incorporating PROMs curtailed classifier performance (accuracy: 68.67%, sensitivity: 91.23%, specificity: 18.52%, F1: .800).

## Conclusion

This study is the first to demonstrate automated clinical classification of NSLBP using computer vision and machine learning with standard video data, achieving accuracy comparable to expert consensus. Automated classification of NSLBP based on altered movement patters video-recorded during routine clinical examination could expedite personalised NSLBP rehabilitation management, circumventing existing healthcare constraints. This advancement holds significant promise for patients and healthcare services alike.

## Introduction

Low back pain (LBP) stands as the most prevalent musculoskeletal ailment globally [1]. It leads to impaired activity and absenteeism, incurring substantial economic consequences [2]. A matter of concern is the escalating growth of disability related to LBP, with the number of years lived with LBP-associated disability surging from 42.5 million in 1990 to 64.9 million in 2017, making LBP the leading global cause of disability [3]. Consequently, LBP has evolved into a prominent global public health concern [4].

Approximately 80–90% of LBP falls under the category of 'non-specific' (NSLBP), lacking identifiable pathology [5]. Management focuses on reducing pain and disability through active physical approaches involving education, self-management, and exercise tailored to individual needs and capabilities [6]. However, tailoring NSLBP management becomes intricate due to the recognised complexity and heterogenous nature of NSLBP including diverse posture and movement alterations [7–10], and varying physical conditioning status and disability [11]. This presents a significant clinical hurdle.

Multiple clinical classification systems have emerged to tailor NSLBP management [12–14]. Among these, substantial evidence and clinical endorsement exist for multidimensional classification (MDC) framework [13]. The MDC is a multi-stage process, described in detail elsewhere [13, 15], to primarily determine the relative dominance of physical, psychological, behavioural and lifestyle factors and its role in perpetuating the NSLBP disorder to better target management. Individuals whose altered movement and posture patterns contribute to their pain, are classified into either (i) *movement impairment (MI)*, characterised by restricted movement due to pain and avoidance behaviour, or (ii) *motor control impairment (MCI)*, characterised by unrestricted yet painful movement and pain-provoking behaviour [13]. NSLBP patients with MI and MCI, directly amendable through exercise interventions, are believed to represent 60% of all NSLBP cases with MI and MCIs manifesting at different spine levels and movement directions (flexion, extension, side-flexion, rotation), and can also be multi-directional [13].

The distinct nature of the subgroup-specific spine movement and posture alterations was extensively studied and was demonstrated across different NSLBP subgroups during sitting

and standing [8, 16–19], spine flexion and extension [20, 21] and functional tasks [22]. Further, MDC based targeted NSLBP management, called Cognitive Functional Therapy (CBT), has growing body of high quality evidence of achieving significant and clinically important improvements in patient outcomes compared to standard practice [23–26].

Clinical delivery of tailored NSLBP management directly depends on clinicians' ability classify. Research shows that trained health professionals demonstrate substantial inter-examiner agreement when classifying NSLBP using MDC [15, 27], however, the agreement appears to depend on the level of training received with agreement dropping from Kappa of .90 in those receiving > 100 hours of training to Kappa of .66 in those with < 100 hours of training [15]. Therefore, despite the compelling evidence of its potential clinical benefits, the resource intensive and time-consuming nature of the classification process and the amount of training required makes delivery of tailored NSLBP management immensely challenging [28]. This poses a challenge to widespread clinical adoption.

Machine learning (ML), a subset of artificial intelligence (AI), employs data and algorithms to discern patterns akin to human intelligence. In the medical sphere, ML is increasingly applied for prognosis, diagnosis, and personalised treatments [29]. It has also found utility in diagnosing and prognosticating LBP, though its precision lags behind clinical classification [30]. AI-based classifiers have emerged in studies classifying NSLBP using 3D-motion capture data [18] and wearable sensor data [9, 31]. While their accuracy holds promise, the added equipment costs, maintenance, and significant computational workload impede clinical integration within current healthcare systems.

Computer vision, a facet of AI, extracts profound insights from digital or video imagery. Clinicians routinely use digital images and videos to categorise altered posture and movement patterns associated with NSLBP [8, 15, 27]. Recent strides in computer vision, including convolutional neural networks (CNNs) and expansive training datasets like COCO [32], enabled human pose estimation through video imagery during gait for example [33]. However to our knowledge, no studies have pursued such techniques for NSLBP classification from digital images or video.

In this study, our objective was twofold: Firstly, to construct and evaluate performance of an ML model for NSLBP classification, we called BACK-to-MOVE, to differentiate between the MI and MCI classes using standard video recordings of routine physical examination of NSLBP patients who were clinically classified as either MI or MCI. Given NSLBP's multifaceted nature, our secondary objective encompassed evaluating the model's performance in tandem with customary patient-reported outcome measures (PROMs) regularly employed to target NSLBP management.

## Methods

The overall protocol utilised in this study consisted of data collection and expert classification of the dataset, pose estimation model design, feature extraction and selection, model training and evaluation described in detail in sections below.

### Participants

We used a patient dataset derived from a group of 83 individuals (n = 47 females; mean age [SD] 44.7 [11.8] years; height 170 [9.9] centimetres; mass 81.3 [6.7] kilograms) experiencing NSLBP for over three months. Recruitment was carried out from the physiotherapy waiting lists at Cardiff and Vale Orthopaedic Centre. Eligibility screening was conducted via both telephone and in-person assessments by an experienced physiotherapy clinician (LS), ensuring adherence to inclusion and exclusion criteria (Table 1). Participants attended a single

**Table 1. Eligibility criteria for NSLBP.**

**Inclusion Criteria**

- Age 18 years +

- Non-specific low back pain for longer than 3 months

- Clear mechanical basis for low back pain whereby symptoms are eased and provoked by specific postures or movements based on previously published criteria [13].
  - Movement impairment (MI)–impairment or a *deficit is in the movement restriction of the symptomatic spinal level*. Pain is triggered by movement of the spine with the movement restricted.
  - Motor control impairment (MCI)—impairment or *deficit in the control* of the *symptomatic spinal level*. Pain is triggered by adopting sustained positions such as sitting, bending, standing, twisting, with movement not restricted.

- Able to speak and understand English well enough to complete questionnaires independently

**Exclusion Criteria**

- Primary pain area different to lower back (from Thoracic vertebra T12 to buttock line) e.g., leg or neck pain

- Acute exacerbation of pain at the time of testing rendering the individual unable to undertake the testing procedure

- Specific diagnosis for pain (nerve root compression, radicular pain/radiculopathy, disc herniation, spondylolisthesis, spinal stenosis)

- Surgery (lower limb or abdominal surgery in last 6 months, any spinal surgery)

- Injection therapy for pain relief in the last 3 months

- Rheumatologic/inflammatory disease (e.g., psoriatic arthritis, rheumatoid arthritis, ankylosing spondylitis, Scheuermann's disease) Scoliosis (if a primary pain driver)

- Progressive neurological or neurodegenerative conditions (e.g. multiple sclerosis, Parkinson's disease, motor neuron disease)

- Red flags/serious pathology (malignancy, acute trauma such as fracture, systemic infection, spinal cord compression, cauda equina syndrome.

- Pregnancy/breast feeding

experimental session at the Research Centre for Clinical Kinaesiology, School of Healthcare Sciences, Cardiff University.

Data used in this study were collected during a Postdoctoral Research Fellowship (LS) between 1 February 2018 to 31 January 2019 with the archived samples accessed 1 June 2020 to conduct this current study. Health and Care Research Wales Ethics Committee, Wales REC 3 (reference 10/MRE09/28, IRAS project number 51853, amendment number ARUKBBC2017.02) approved the study. Participant anonymity was ensured by applying facial blurring to each frame of the video files, executed by the postdoctoral researcher (TH) using a custom Matlab script. All subsequent data processing and analysis was performed on a fully anonymised dataset.

## Clinical classification procedure

Each enrolled NSLBP participant underwent classification utilising the clinical MDC framework [13]. This classification was independently executed by two expert physiotherapists, both well-versed in MDC and trained in accordance with a documented classification protocol [15]. This process entails a subjective evaluation of pain behaviour, scrutiny of available radiological imaging, assessment of pain-related beliefs, lifestyle habits, and physical examination of spinal function. The latter includes gauging the range, speed of movement, and pain response during fundamental movements such as spine flexion, extension and side flexion as well as evaluating

slouched and upright sitting postures, and basic functional tasks like sit-to-stand and squat. A clinical judgment is then rendered regarding whether observed movement irregularities are the dominant pain drivers. Subsequently, individuals were broadly categorised into either MI (n = 42) or MCI (n = 41) phenotypes, based on the criteria outlined in Table 1. and independently confirmed by two expert physiotherapists.

## Data acquisition

The patient dataset incorporated movement data from a battery of physical assessment of the spinal function informed by expert clinical consensus as integral to NSLBP classification [34]. These assessments encompassed spine flexion, extension, side flexion, along with functional tasks like sit-to-stand, squat and both upright and slouched sitting postures. From this assessment battery, the clinical importance of spine flexion demonstrated the highest expert consensus at 98% [34]. Consequently, spine flexion was chosen as the initial movement test for predicting NSLBP MI and MCI classes in this study. Each participant performed 8–10 repetitions of spine flexion, preceded by two practice repetitions for warm-up. Participants were instructed to begin from a relaxed standing position and bend forward as much as possible, with no further guidance on the method or speed of the movement.

Both an optoelectronic motion capture system (Vicon, Oxford, UK) and a digital video camera (Hero 5, GoPro Inc., CA, USA) captured movement data simultaneously (Fig 1). For motion capture system data, retroreflective markers were positioned on the seventh cervical vertebra (C7), lumbar vertebra (L4), and pelvis (anterior and posterior iliac spines—ASIS, PSIS). Inertial measurement units (not discussed here) attached to the spine necessitated positioning the L4 marker parallel to the fourth lumbar vertebra. Retroreflective marker positions were recorded at 120 Hz using Vicon MX cameras and Vicon Nexus software.

All videos were pre-processed by undergoing rescaling to ensure the shortest side was 768 pixels in length and were resampled for 30 frames per second playback using open-source Python and OpenCV software packages.

## Patient-reported outcome measures

Prior to engaging in any physical movement tests, participants were requested to complete PROMs validated and routinely used in LBP to profile pain intensity, levels of disability, fear of

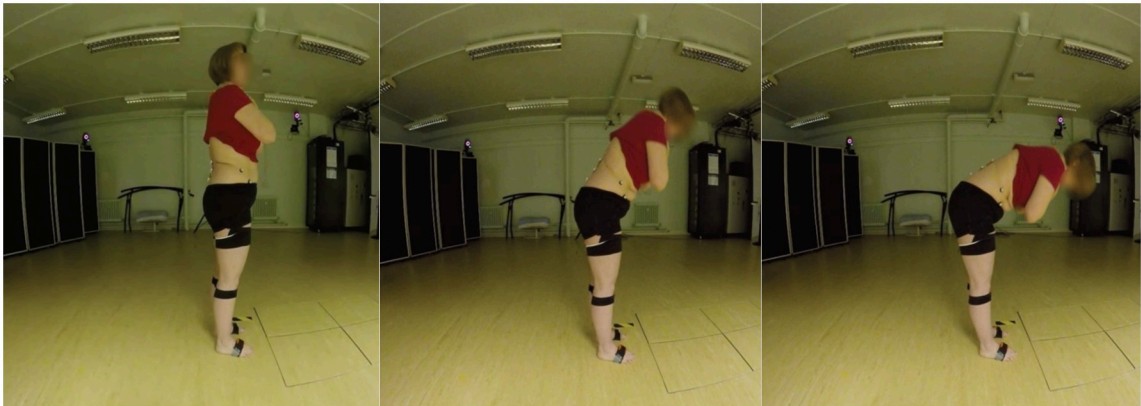

**Fig 1. Experimental set up for obtaining spine flexion.**

**Table 2. Patient-reported outcome measures (PROMs).**

| Measure | Domain | Description |
|---|---|---|
| Visual Analog Scale (VAS) [35] | Pain Intensity | Level of pain marked on a 100-mm scale anchored with "No pain" and "Worst imaginable pain" at the respective ends. |
| Oswestry Disability Index (ODI) [36] | Disability | Participants asked to tick statements that best applying to them in 8 activity of daily living domains (e.g., lifting, personal care and walking). This index's score spans from zero to 100, with higher scores signifying more pronounced disability. |
| Tampa Scale of Kinesiophobia [37] | Fear of movement | Measure of movement-related apprehension in LBP individuals. The scale's score range is 17 to 68, with elevated scores reflecting greater kinesiophobia. |
| Pain Catastrophising Scale (PCS) [38] | Pain catastrophising | 13 statements describing thoughts associated with pain, scored using a 5-point scale anchored with 'not at all' and 'All the time' at the respective ends. |
| Pain Self-efficacy Questionnaire (PSEQ) [39] | Pain Self-efficacy | 10 statements describing activities of daily life scored on a 6-point scale anchored with 'Not at all confident' to 'Completely confident' at the respective ends. |
| Coping Strategies Questionnaire (CSQ) [40] | Coping | A questionnaire asking participants to score 14 items describing different ways of coping with pain using a 6-point scale anchored with 'Never do that' to 'Always do that' at the respective ends. |
| STarTBacK tool (SBT) [41] | Risk of developing persistent LBP-related disability | 9-item screening questionnaire to stratify LBP patients based on their risk (low, medium, high) for persistent LBP-related disability. |

movement, pain related catastrophising, self-efficacy, coping strategies, and level of risk of developing persistent LBP disability (Table 2).

## Human pose estimation

Convolutional Neural Networks (CNNs) [42]—a subset of artificial neural networks—were harnessed to deduce human body posture from visual data, enabling the phenotyping of NSLBP. Multiple CNN architectures tailored for human pose estimation were considered: CrowdPose [43], DensePose [44], PoseNet [45], UniPose [46] and HigherHRNet [47]. After initial exploration, the HigherHRNet model was chosen due to its exceptional performance, effectively accommodating the varied forward bending movement characteristics inherent in the patient dataset. A detailed comparison of HigherHRNet with alternative CNN architectures was presented in the original article clearly demonstrating its superiority in accurate estimation of the human pose in the videos [47], which was the main influence of the choice.

HigherHRNet, operating as a bottom-up pose estimation model, directly infers human body posture by identifying key body points (keypoints) throughout the entire image [47]. In the initial phase, all 17 keypoints within the HigherHRNet model were identified across the video data of each participant. However, for classifying this NSLBP patient dataset, only specific keypoints—the ankle, hip, and neck, as illustrated in Fig 2, were deemed essential to capture clinically pertinent features essential for classification. The x, y coordinates of each identified keypoint were recorded in pixel space.

By identifying keypoints across all frames within each video, a movement pattern of the participants was established. This concept is exemplified in Fig 3, depicting the x, y coordinates, representing anteroposterior and vertical directions, respectively, of the keypoint representing the neck for a single participant executing multiple repetitions of spine flexion.

The initial pre-trained human pose estimation model, HigherHRNet, was trained on the MS-COCO dataset containing 250,000 person instances distributed across 200,000 images [32]. Fine-tuning the pre-trained model wasn't pursued, as it already exhibited strong performance. This decision aimed to prevent excessive overfitting to the training data, which might compromise the model's capacity to encapsulate the diverse movement nuances present in the dataset [48].

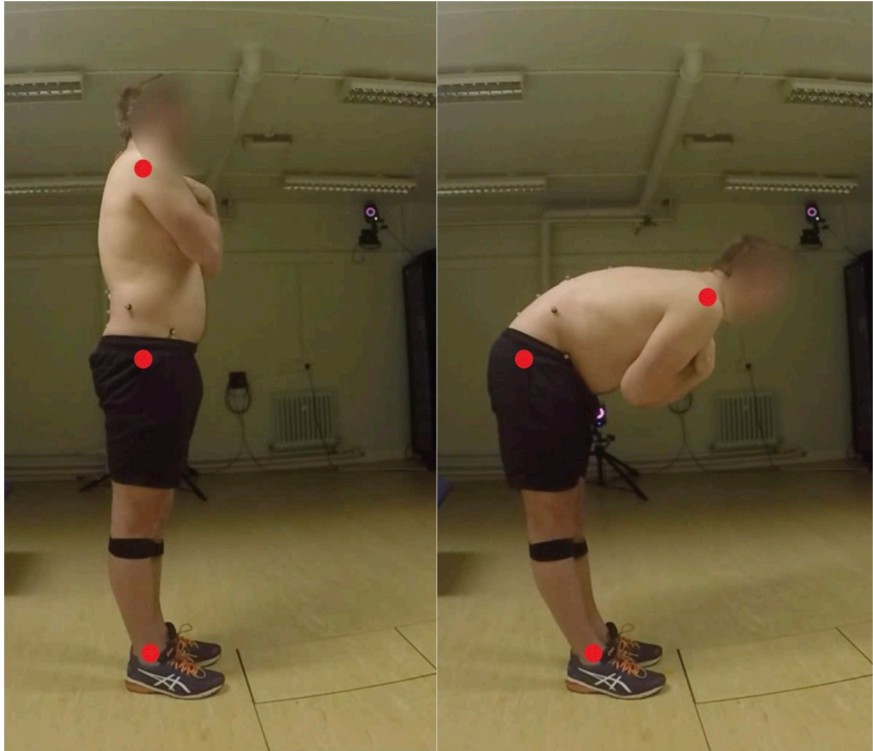

**Fig 2. Human pose estimation model utilised to estimate spine flexion performance.**

## Feature extraction

In this study, the selected features were deliberately aligned with what holds significance for classifying NSLBP in a clinical context. This approach aimed to avoid overfitting, a well-recognised challenge associated with constructing classifiers that incorporate numerous high-dimensional features [49]. Among the most prevalent NSLBP characteristics are impairments in spatial and temporal parameters of range of movement [7] a focal point of attention during NSLBP assessments undertaken by physiotherapists.

Range of movement was quantified in the form of a spinal angle, computed as the Euclidean distances (in pixels) between three points derived from the pose estimation process. These distances encompass those between the hip and neck keypoints (hn), ankle and hip keypoints (ah), and ankle and neck keypoints (an), as visualised in Fig 4. Armed with these distances, the angle of the spine flexion (θ) was determined using the following formula:

$$\theta = \arccos(hn^2 + ah^2 - an^2)/(2 \cdot (hn \cdot ah))$$

Subsequently, a moving average with a 30-frame window (equivalent to 1 second of video data) was employed to smoothen the spine flexion angle waveform. This process aimed to mitigate any unforeseen data fluctuations (Fig 5).

A collection of both established and new statistical features, commonly applied in clinical movement and NSLBP classification analyses [8, 20, 50], were then computed for the angle signal. These features were employed to assess their predictive value for distinguishing MI and MCI classes within NSLBP. Among the standard features routinely used in clinical movement

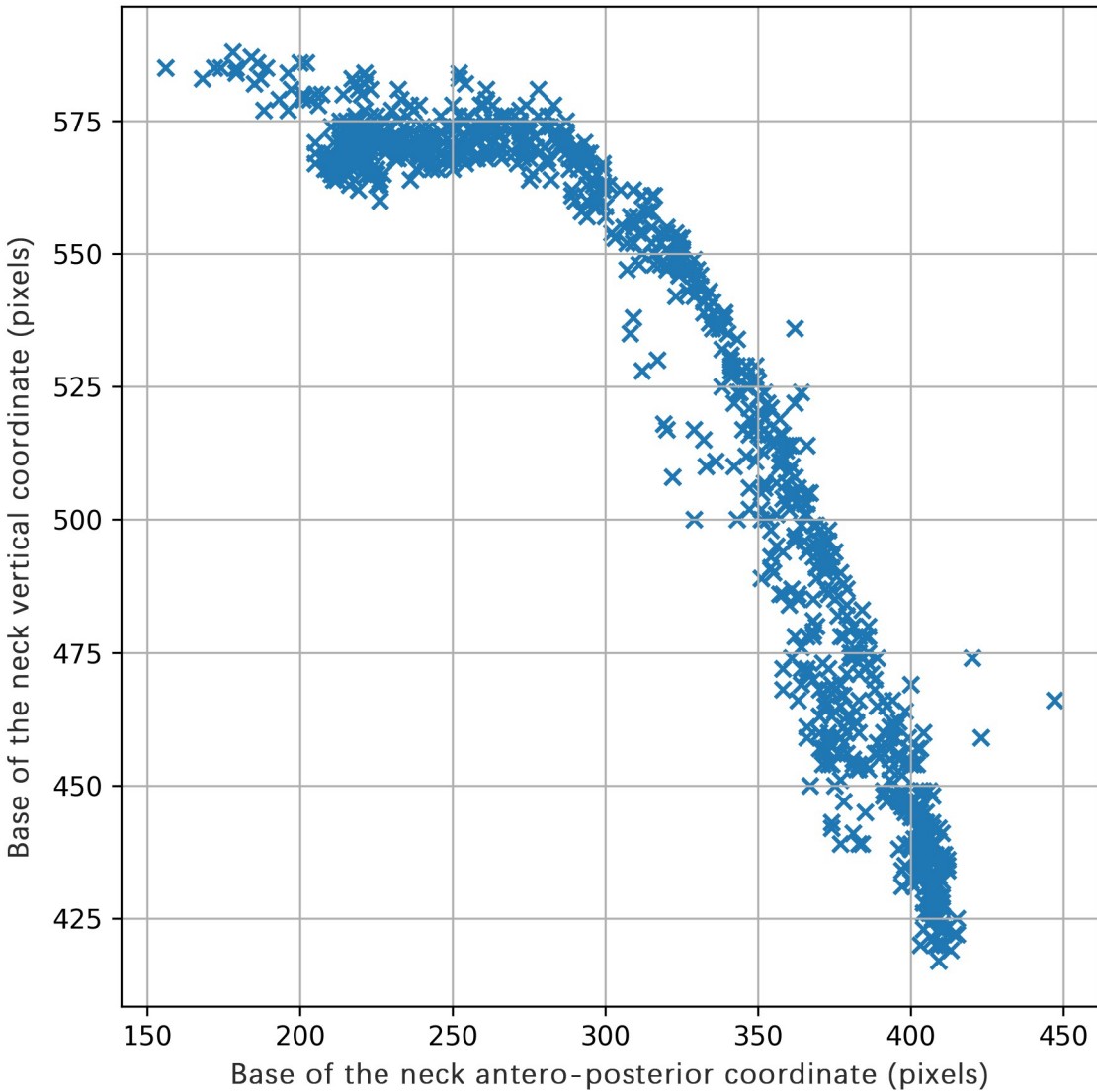

**Fig 3. Sagittal 2D position of the base of the neck keypoint.** Y = base of the neck vertical coordinate (pixels), X = base of the neck antero-posterior coordinate (pixels).

analysis were the mean, range, minimum, maximum values, variance, and standard deviation for the spine flexion angle θ (Fig 4).

The new features introduced here aimed to evaluate the temporal and spatial consistency of the spinal flexion movement. To achieve this, the angle waveform was scrutinised (Fig 5), with particular emphasis on its minima points, which hold clinical significance as they correspond to the points when participants reach the lowest flexion (Figs 4 and 5). The spine flexion repetition time ($tr_i$) was deduced from the frame count between these minima points on the angle graph. Detecting these minima points was accomplished through signal inversion coupled with a peak detection algorithm, in this instance, the approach by Du et al. (2006) [51], implemented through the SciPy python library's find_peaks_cwt function.

Novel features derived from these minima points encompassed the mean and variance of repetition time ($tr_i$) and the variance of the spine flexion angle depth ($depth_i$) in Fig 5, further

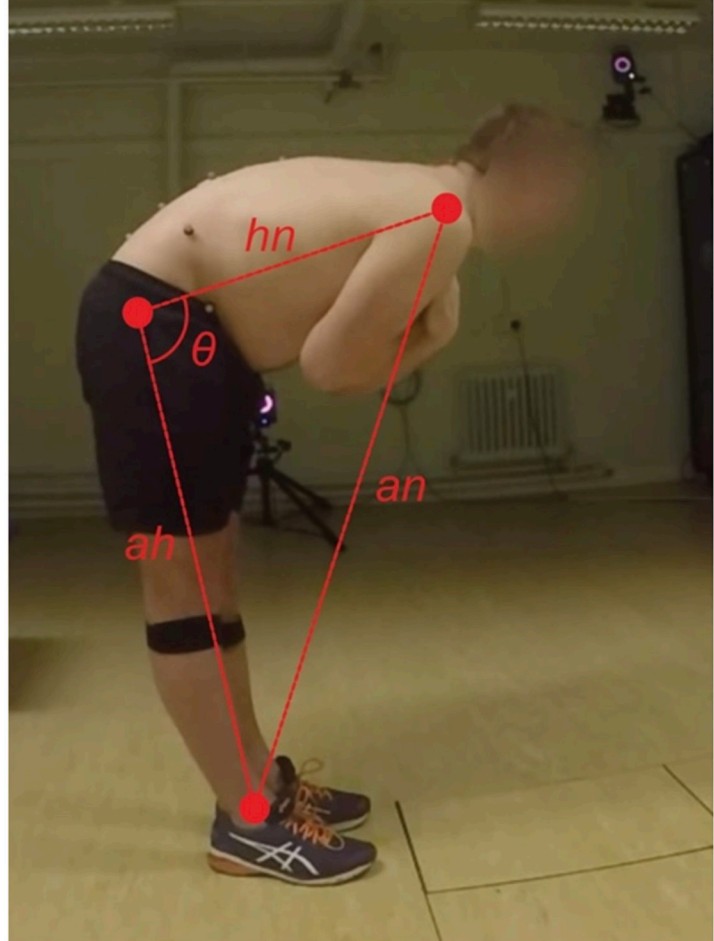

**Fig 4. Visual representation of the calculation to obtain the range of movement of spine flexion.** Euclidean distance (in pixels) between hip to neck (hn), ankle to hip (ah), and ankle to neck (an).

elaborated below. The variance in depth ($depth_i$) provided insight into the consistency of participants attainment of a specific flexion depth and the extent to which the flexion depth fluctuated across repetitions. Lastly, movement stability, the final feature extracted from the angle graphs, was obtained by dividing the angle waveform (Fig 5) into two halves and determining the angle range for each half, then subtracting one from the other. This feature offered insight into whether a patient increased their maximum bend angle across several repetitions [51].

Considering the multifaceted character of NSLBP, the collected PROMs were incorporated as other important clinical features. This culminated in a set of 16 features for distinguishing between NSLBP MI and MCI subsets (Table 3).

## Predictive model design

Category prediction was executed using feedforward neural network, with parameter values drawn from a training dataset comprising labelled instances [52]. The predictive model adopted here (Fig 6) encompassed multiple layers, each featuring a distinct type of input–angle and PROMs (Table 3). These inputs were processed through individual streams, which first underwent batch normalisation [53] before progressing to a feedforward layer. This

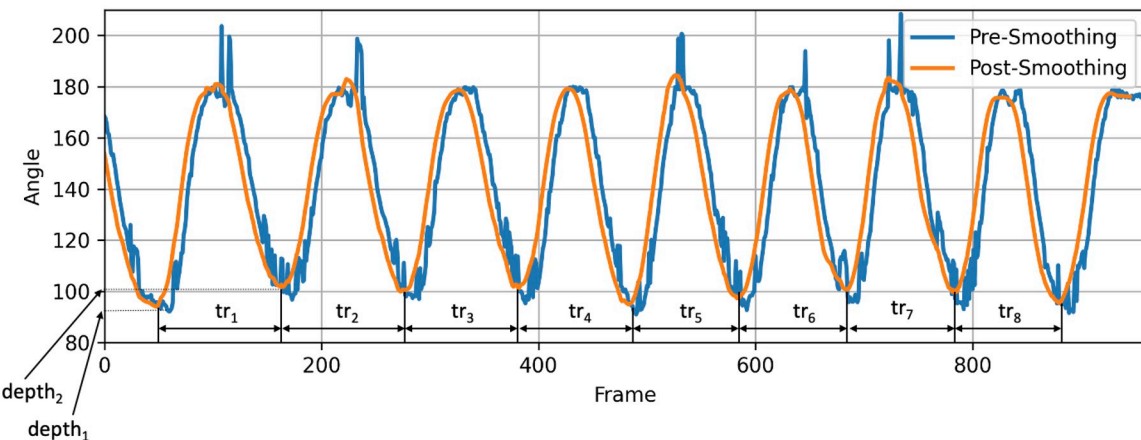

**Fig 5. Spine flexion angle waveform of a single participant repeating the movement with the signal pre-smoothing (blue) and post-smoothing (orange).** The spine flexion repetition time ($tr_i$), and angle depth ($depth_i$) are highlighted in the graph to give insight into the consistency of participants attainment of a specific flexion depth and the extent to which the flexion depth fluctuated across repetitions.

feedforward layer entailed no hidden nodes; input and output nodes equated the number of features. A Rectified Linear Unit (ReLU) activation function followed the linear layer [54]. The streams were then concatenated and fed into a concluding feedforward layer equipped with a threshold, yielding a binary output (0 or 1) signifying the prediction of either MI or MCI. Throughout this process, model was trained using binary cross-entropy loss and stochastic gradient descent, following a supervised methodology [55].

**Table 3. Movement and clinical features used for distinguishing between MI and MCI.**

| Feature Domain | | Input features |
|---|---|---|
| **Spine Flexion Angle** | *Standard* | 1. Variance |
| | | 2. Standard Deviation |
| | | 3. Minimal (full spine flexion angle) |
| | | 4. Range |
| | | 5. Maximal (spine standing angle) |
| | *Novel* | 6. Repetition Time Mean |
| | | 7. Repetition Time Variance |
| | | 8. Depth Variance |
| | | 9. Movement Stability |
| **Patient Reported Outcome Measures (PROMs)** | | 10. Visual Analog Scale (VAS) |
| | | 11. Oswestry Disability index (ODI) |
| | | 12. Tampa Scale of Kinesiophobia (TSK) |
| | | 13. Pain Catastrophising Scale (PCS) |
| | | 14. Patient Self-efficacy Questionnaire (PSEQ) |
| | | 15. Coping Strategies Questionnaire (CSQ) |
| | | 16. STarTBacK tool (SBT) |

(MI = movement impairment, MCI = motor control impairment)

**Fig 6. Predictive model.** Batch Norm = batch normalisation layer, ReLU = Rectified Linear Unit layer.

## Evaluation

**Pose estimation validation.** Criterion validity of the pose estimation model was assessed by comparing the spine flexion angle acquired through pose estimation with the synchronised 'ground truth' 3-dimensional (3-D) motion capture (Mocap) data, utilising 75% of the NSLBP dataset (n = 62). In the case of the 3-D Mocap spine flexion angle, the pose estimation keypoints (neck and hip) were replaced with the corresponding Mocap positions of C7 and a midpoint between the pelvic markers (PSIS and ASIS). Approximating the ankle's position involved setting the y position of the L4 marker to zero, generating a virtual ankle marker (vA) consistently positioned directly beneath the L4 marker at a vertical position of zero. The spine flexion angle calculation was then executed as detailed in section on 'Feature extraction' above. Subsequently, the pose's mean square error score was computed to quantify the disparity between the spine flexion angle extrapolated from the Mocap data and that deduced from the pose estimation keypoints.

**Predictive model evaluation.** The evaluation of the predictive model involved two aspects: i) selecting the most pertinent features and ii) assessing the final model's performance. To appraise the predictive model's performance, classification accuracy, along with sensitivity, specificity, and the F1 measure, were chosen as evaluation metrics. Accuracy signifies the fraction of predictions the model correctly attributed. Sensitivity and specificity offer insights into the nature of misclassifications.

In the context of a binary predictive model, as in this study, sensitivity denotes the model's ability to accurately identify all instances of the positive class (MI), while specificity gauges the model's capacity to correctly recognise all instances of the negative class (MCI). These metrics were computed employing true positive detections (TP), true negative detections (TN), false positive detections (FP), and false negative detections (FN), using the subsequent formulae:

$$Accuracy = \frac{TP + TN}{TP + FP + TN + FN}$$

$$Sensitivity = \frac{TP}{TP + FN}$$

$$Specificity = \frac{TN}{TN + FP}$$

To gauge the overall classification performance of the predictive model, the F1 measure was computed. The F1 measure represents a balanced combination of precision (sensitivity) and recall (ability to detect class samples), ranging from 0 to 1, where 0 signifies the lowest and 1 indicates the highest overall classification performance [56].

The model assessment occurred in three progressive stages, with outcomes from each stage guiding the subsequent one. The initial stage involved exploring the significance of various

feature types in accurate predictions. To achieve this, the model underwent training using all available features, followed by training with a single type of feature. The resulting metric values served as baseline for comparison with subsequent experiment results. In the second stage, the model's performance was assessed for all permutations of each feature type. Angle features presented 512 permutations [29], while PROM features featured 128 permutations [27]. The final stage encompassed constructing an optimal feature set, in the sense of producing the highest F1-measure among all possible permutations of features in the corresponding model assessment, finalising model training, and executing an ablation experiment to analyse each feature's impact on the final model's accuracy. All assessment measures were calculated based on 5-fold cross-validation.

Statistical significance of the results can be evaluated when comparing various competing classification models; however, there is no prior work against which our proposed model could be benchmarked. Nonetheless, the statistical significance of the accuracy obtained in this study was assessed following the method outlined in Combrisson et al., 2015 [57] suitable for comparisons of comparable sample size. For $c$ classes and an infinite number of equally distributed samples, the percent theoretical chance level of classification is $100/c$. In our case, the number of classes $c = 2$, and the chance level of classification is 50%. However, for a limited number of samples, the chance level will depend on it. As Combrisson et al, we assume that the classification errors obey a binomial cumulative distribution, where the probability to predict the correct class at least $z$ times by chance is given by:

$$P(z) = \sum_{i=z}^{n} \binom{n}{i} \times \left(\frac{1}{c}\right)^i \times \left(\frac{c-1}{c}\right)^{n-1}$$

Following this, we used the MATLAB (Mathworks Inc., MA, USA) code provided by Combrisson et al (2015) to compute the statistically significant threshold $St(\alpha) = binoinv(1-\alpha, n, 1/c) \times 100/n$, where $n$ is the sample size, $c$ is the number of classes, $\alpha = z/n$ is the significance level and $binoinv()$ is a MATLAB function [57].

## Results

### Pose estimation model validity

An illustration of an angle waveform obtained from the pose estimation model and a waveform derived from motion capture data for a single participant performing spine flexion is presented in Fig 7. The mean square error score, calculated to quantify the divergence between the two data sets, was 0.35 degrees. This result showcases robust criterion validity and an adequate level of accuracy between the pose estimation model and the motion capture data.

### Predictive model evaluation

The classification outcomes from the initial stage of experiments involving all angle and all PROMs input features are presented in Table 4. The higher level of accuracy, sensitivity, and specificity was attained by employing 'all angle features'. The PROMs demonstrated relatively weak capability to precisely identify MCI, resulting in a reduced overall accuracy and implying limited contribution to accurately differentiate between the two NSLBP subsets (Table 4).

Table 5 presents a set of eight optimal input features (marked by X) that was curated from the relevant 16 features on basis of yielding the highest F1 measure value in correctly discriminating between MI and MCI. Interestingly, standard features employed in clinical movement analysis (angle minimum and variance) did not exhibit utility in distinguishing between the

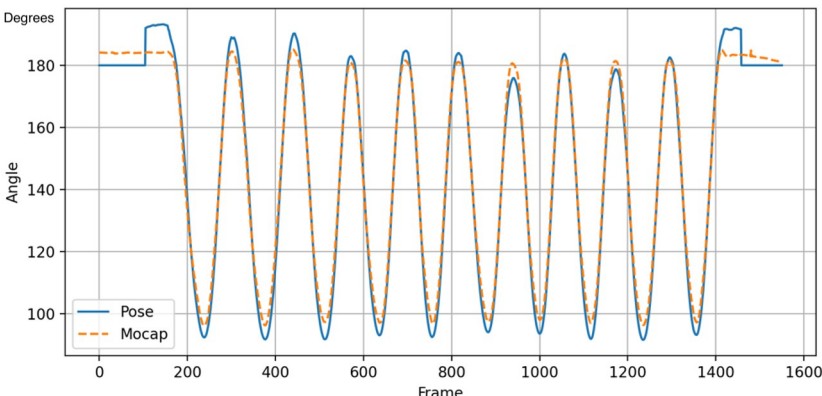

**Fig 7. An example of an angle waveform from a single participant performing 10 repetitions of spine flexion derived from the pose estimation model and the motion capture data (Mocap).** Blue = pose estimation model; Orange = Mocap.

subgroups and STarTBacK Tool, a measure of risk in developing disabling LBP, was the only PROM feature selected (Table 5).

Lastly, the classification performance using solely the optimal input features, both individually and in combination, is depicted in Table 6. The highest overall classification performance of 93.98% was achieved when exclusively utilising optimal angle features. The addition of optimal PROMs feature to the optimal angle features did not yield improvement in the classification performance.

The above results were subjected to statistical testing described earlier in the article, achieving statistical significance (p < 0.001) for accuracies of 87.5% and above. Therefore, our results are confirmed to be statistically significant.

## Discussion

This is a novel study that examines the feasibility of BACK-to-MOVE; a machine learning and computer vision model for automating clinical classification of NSLBP on basis of altered spine movement captured on standard video camera. A recent systematic review revealed that computer vision's primary use in LBP predominantly revolves around extracting features from radiological imaging including MRI, CT, X-ray, or ultrasound to identify and/or categorise vertebral structures and intervertebral discs' pathology levels [30]. While these models may demonstrate clinical applications for specific LBP with a clear patho-anatomical cause, it's important to note that over 80% of LBP cases are non-specific with clinical guidance actively discouraging the use of imaging to guide its management. [58]. This study stands as the first to

**Table 4. Classification performance when all spine flexion angle and patient reported outcome measures (PROMs) features were inputted.**

| Features | Accuracy | Sensitivity* | Specificity** | F1 Score |
|---|---|---|---|---|
| Spine Flexion Angle (all) | 87.95% | 92.98% | 76.92% | .914 |
| PROMs (all) | 68.67% | 91.23% | 18.52% | .800 |

*Sensitivity denotes accuracy of correctly assigning Movement Impairment.

**Specificity denotes accuracy of correctly assigning Motor Control Impairment.

F1 = overall classification performance, 0 = lowest, 1 = highest.

**Table 5. Set of optimal input features yielding the highest classification performance.**

| Feature domain | | Input features | Optimal features marked by X |
|---|---|---|---|
| **Spine Flexion Angle** | *Standard* | 1. Variance | - |
| | | 2. Standard Deviation | X |
| | | 3. Minimal (full spine flexion angle) | - |
| | | 4. Range | X |
| | | 5. Maximal (spine standing angle) | X |
| | *Novel* | 6. Repetition Time Mean | X |
| | | 7. Repetition Time Variance | X |
| | | 8. Depth Variance | X |
| | | 9. Movement Stability | X |
| **Patient Reported Outcome Measures (PROMs)** | | 10. Visual Analog Scale (VAS) | - |
| | | 11. Oswestry Disability index (ODI) | - |
| | | 12. Tampa Scale of Kinesiophobia (TSK) | - |
| | | 13. Pain Catastrophising Scale (PCS) | - |
| | | 14. Patient Self-efficacy Questionnaire (PSEQ) | - |
| | | 15. Coping Strategies Questionnaire (CSQ) | - |
| | | 16. STarTBacK tool (SBT) | X |

X = features achieving highest F1 measure value among all possible permutations of features

employ AI in classifying *non-specific* LBP utilising video imagery with the intent to aid tailored management of NSLBP as endorsed by clinical guidelines [6]. This approach therefore holds a significant potential to enhance the clinical viability and usefulness of such methods.

This study BACK-to-MOVE classifier achieved percentage accuracy of 93.98% in predicting MI and MCI with F1 score of 0.957, signifying exceptional performance. This level of accuracy favourably compares to inter-examiner agreement of highly experienced physiotherapists undergoing 100+ hours of training to achieve a Kappa coefficient of .82 and percentage agreement of 86%, and surpasses inter-examiner agreement of less experienced clinicians (undergoing less than 100 hours training) achieving Kappa of 0.66 [15]. Our classifier also outperformed clinicians classifying NSLBP from videos of physical examination to reach a Kappa between .55 and .71, and percentage agreement between 65 and 78% [27].

The BACK-to-MOVE classifier showed better prediction of MI class (96.49% sensitivity) than MCI class (88.46% specificity). This difference could stem from the fact that MI, characterised by spine movement restriction, is perhaps more easily discernible from video compared to the potentially subtler deviations in motor control associated with MCI class [10, 13].

**Table 6. Classification performance when the optimal features were inputted separately and combined.**

| Features | Accuracy | Sensitivity* | Specificity** | F1 Score |
|---|---|---|---|---|
| Optimal Spine Flexion Angle | 93.98% | 96.49% | 88.46% | .957 |
| Optimal PROMs | 73.49% | 91.23% | 34.62% | .825 |
| Optimal Spine Flexion Angle + PROMs | 92.77% | 94.74% | 88.46% | .947 |

*Sensitivity denotes accuracy of correctly assigning Movement Impairment.

**Specificity denotes accuracy of correctly assigning Motor Control Impairment.

F1 = overall classification performance, 0 = lowest, 1 = highest

Interestingly, Fersum et al. (2009) demonstrated higher clinical agreement classifying MCI (99% agreement) than MI (75% agreement) [15]. This incongruity might be attributed to the uneven distribution of cases in the Fersum et al (2009) study with 24 cases of MCI and one case of MI, along with the clinicians' suggested unfamiliarity with the less frequently encountered MI [13, 15]. In our study, an almost equal numbers of cases were included across MI (n = 41) and MCI (n = 42).

The highest accuracy (93.98%) was achieved by using seven optimal movement features that characterise the spatial and temporal parameters of spine flexion, extracted from the video imagery. This is in line with recent systematic review and meta-analysis demonstrating that people with LBP have impairments in movement range, speed and spine flexion variability [59]. The advantage of these movement features likely arises from their objective nature encompassing statistical properties of the spine flexion angle's behaviour across the movement repetitions. Among the 9 movement features, 2 (variance and minimum angle value) were not chosen by the model. This could be because they were already represented by the remaining features, specifically, standard deviation, range, and maximum.

To address the multifactorial nature of NSLBP, we examined the inclusion of seven patient-reported outcome measures (PROMs) typically considered during LBP classification including levels of pain, disability, fear of movement, catastrophising, coping, self-efficacy and risk of chronicity. Using all these PROMs as input features alone achieved low accuracy of 68.67%, rising marginally to 73.39% when only the optimal PROMs were used (STarTBacK). Combining optimal PROMs with the optimal movement features enhanced the classification accuracy to 92.77%, however, this was still lower than inputting the optimal movement features alone (93.98%). The limited discriminative capability of PROMs in our study was likely due to the specific recruitment criteria, which focused on participants with LBP persisting for more than three months, while excluding those experiencing acute and severe LBP. This targeted selection criteria likely contributed to the uniformity of PROMs scores, with variations minimised through excluding the acute and severe cases. In summary, while PROMs undoubtedly provide more holistic picture of the patient's pain experience, they appear non-discriminatory when it comes to classifying NSLBP of similar intensity and duration based on physical characteristics thus may not be useful features for ML classifiers helping to discern between altered movement patterns.

The BACK-to-MOVE classifier developed in this study utilised a dataset of 83 NSLBP patients, a sample size comparable and exceeding many other studies [60]. To maintain the dataset's full complexity and minimise selection bias, we included video data from all participants, regardless of challenges like lighting variations. This approach reduces the risk of overestimating predictions by avoiding oversimplification of classification scenarios, an aspect highlighted in previous research [61]. However, some selection bias was unavoidable as we excluded two participant sets with videos too brief for clinicians to classify from. It's also important to note that participants in our dataset were wearing sports clothing, which could have positively affected pose estimation accuracy. Further exploration is warranted to assess the extent of this influence. Lastly, while participants in our dataset had Vicon markers attached to their spine and pelvis, the model used in this study was trained on the standard COCO dataset, which does not contain any markers. This disparity is therefore unlikely to have significantly impacted pose estimation accuracy.

## Strengths and limitations

This study's primary strength lies in highly innovative use of the state-of-the-art human pose estimation model, HigherHRNet, to classify NSLBP according to a clinical classification model,

achieving an accuracy exceeding 93%, which is comparable to clinical expert agreement. Although various pose estimation algorithms and models exist, none have proven reliable for estimating the entire spine's pose. Therefore, we focused on measuring a single spine flexion angle, which proved effective. However, this single-angle focus might be a limitation considering the complexity of NSLBP demonstrating movement and motor control alterations [7, 8, 10].

Another strength is our use of data from a standard video camera alone. Most clinical research relies on specialised motion capture hardware, not easily accessible in standard clinical settings. Our approach offers an alternative path for potential clinical adoption without high costs or space requirements. Nonetheless, this method's accuracy depends on video quality and can be influenced by lighting and background conditions.

We utilised movement features already considered important by clinicians trained in NSLBP classification [34]. This will enable easy interpretation by clinicians and immediate opportunities for existing interventions to be tailored to patient needs and capabilities in accordance current clinical guidelines [62]. We also confirmed that these clinician-considered movement features achieve highly accurate automatic NSLBP classification.

A limitation is evaluating a single task (spine flexion) when considering NSLBP heterogeneity in movement and motor control alteration across different tasks [8, 10, 22]. However, recent data indicate that NSLBP individuals exhibit consistent spinal movement patterns across various tasks, known as a spinal movement signature [7]. While this supports focused data collection, discriminating between directional impairment patterns would likely require data from other movements. This study's feasibility in detecting MI and MCI subsets from a single movement test suggests future research can enhance the automated classifier's sensitivity across other clinical assessment tests, enabling more tailored rehabilitation approaches when and where this is level of specificity is deemed clinically important (e.g. guide specific exercise protocols for NSLBP individuals in higher risk occupations or sports involving spine loading in specific directions).

## Clinical implications

Before considering potential clinical implications, it is important to recognise that classification of a complex and multifactorial such as NSLBP goes beyond physical examination. Other key information including patient history, injury details, radiology (if available), pain behaviour, coping and lifestyle factors are important to identify the dominant pain drivers and target management [24]. Although this study BACK-to-MOVE classifies NSLBP on basis of patient physical function, the primary focus of NSLBP management revolves around restoring physical function [5]. Hence, automating classification of physical aspects is of clinical importance not only to target management but assist its delivery and offer means of objective progression monitoring, thus creating a potential for substantial clinical value to many.

Secondly, NSLBP movement and motor control impairments are described to exhibit directional biases (e.g., extension, flexion, side-flexion, or multi-directional) [13]. In this study, we focused on the initial classification step, determining whether NSLBP primarily involves impaired movement (MI) or impaired motor control (MCI). This distinction enables tailoring the exercise regimen to restore either movement or motor control impairment, as needed.

Self-management tailored to individual needs and capabilities, as endorsed by clinical guidelines, is a cornerstone of managing long-term musculoskeletal pain conditions like NSLBP [6]. Presently, self-management interventions heavily rely on substantial clinical involvement, in-person assessments, and follow-up consultations, with limited avenues for ongoing physical assessment, evaluation, or progress tracking. The BACK-to-MOVE classifier developed in this study offers a significant breakthrough and potential to revolutionise the

delivery of tailored self-management interventions. This encompasses automating clinical movement assessment from videos and presents opportunities like remote condition monitoring and providing prompt progress feedback to patients.

Moreover, BACK-to-MOVE holds the potential to address the challenge of offering timely access to personalised, top-quality rehabilitation for all, aligning with the World Health Organization's call for such advancements [63].

## Future research

Future research directions should focus on overcoming the limitations linked to the simplified model of the human spine employed in this study. With advancements in computer vision models, upcoming investigations will delve into movement patterns spanning the entire spinal curvature, capitalising on the clinically significant movement features examined in this study. This holds the potential to yield fresh insights into the development, progression, and subsequent treatment options for movement disorders in the spine. Another avenue for future research involves leveraging our existing data to explore the directional nature of MI and MCI impairments and their correlation with NSLBP-related disability, thus refining rehabilitation strategies. Lastly, expanding the dataset to include both spinal kinematics and muscle electromyography (EMG) will be helping the model detect non-physiological muscle activation further contributing to crafting more precise and clinically meaningful classification models.

## Conclusion

This study presents the first automated clinical classification model designed for non-specific low back pain, capable of discerning between clinically recognised movement and motor control impairments in the spine from 2D videos, using computer vision and machine learning. Utilising video data alone, BACK-to-MOVE achieved classification accuracy exceeding 93%, aligning to expert clinician consensus, showcasing its potential for practical clinical application. The automation of NSLBP classification through video-recorded altered movement patterns from routine clinical examinations has the potential to enhance the efficiency of personalised NSLBP rehabilitation management, addressing current healthcare limitations. This advancement holds substantial promise for both patients and healthcare services.

## Author Contributions

**Conceptualization:** Yulia Hicks, Jennifer L. Davies, Dario Cazzola, Liba Sheeran.

**Data curation:** Thomas Hartley, Yulia Hicks, Jennifer L. Davies, Dario Cazzola, Liba Sheeran.

**Formal analysis:** Thomas Hartley, Jennifer L. Davies.

**Funding acquisition:** Yulia Hicks, Liba Sheeran.

**Investigation:** Thomas Hartley, Yulia Hicks, Liba Sheeran.

**Methodology:** Yulia Hicks, Jennifer L. Davies, Dario Cazzola, Liba Sheeran.

**Project administration:** Liba Sheeran.

**Resources:** Liba Sheeran.

**Software:** Thomas Hartley.

**Supervision:** Yulia Hicks, Jennifer L. Davies, Dario Cazzola, Liba Sheeran.

**Validation:** Jennifer L. Davies.

**Visualization:** Thomas Hartley, Yulia Hicks, Dario Cazzola, Liba Sheeran.

**Writing – original draft:** Yulia Hicks, Liba Sheeran.

**Writing – review & editing:** Thomas Hartley, Yulia Hicks, Jennifer L. Davies, Dario Cazzola.

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
