## [Decision Letter · Decision Letter 0]

20 Feb 2024

PONE-D-23-28397BACK-to-MOVE: Machine Learning and Computer Vision Model Automating Clinical Classification of Non-Specific Low Back Pain for Personalised Management.PLOS ONE

Dear Dr. Sheeran,

Thank you for submitting your manuscript to PLOS ONE. After careful consideration, we feel that it has merit but does not fully meet PLOS ONE’s publication criteria as it currently stands. Therefore, we invite you to submit a revised version of the manuscript that addresses the points raised during the review process.

We look forward to receiving your revised manuscript.

Kind regards,

Adrian Pranata, Ph.D

Academic Editor

PLOS ONE

Journal Requirements:

Additional Editor Comments:

Thank you for your patience with the review process. This study is a good step forward in the assessment of movement in people with low back pain. Please address the reviewers' comments on the technical aspect of the study's machine learning algorithm. Thank you for your submission.

Reviewers' comments:

Reviewer's Responses to Questions

**Comments to the Author**

1. Is the manuscript technically sound, and do the data support the conclusions?

Reviewer #1: Partly

Reviewer #2: Yes

2. Has the statistical analysis been performed appropriately and rigorously? 

Reviewer #1: No

Reviewer #2: Yes

3. Have the authors made all data underlying the findings in their manuscript fully available?

Reviewer #1: No

Reviewer #2: Yes

4. Is the manuscript presented in an intelligible fashion and written in standard English?

Reviewer #1: Yes

Reviewer #2: Yes

5. Review Comments to the Author

Reviewer #1: The paper has few issues that need further clarifications as follows:

- It is not clear how to validate or ensure data MI and MCI at the beginning before this data fed to feature extraction process.

- The structure of the CNN is not very clear such is the dimension of the input and output.

- Table 5 shows set of optimal input feature yielding the highest classification performance. Instead of using 'X' the actual metric such as classification should be used. Moreover, how to ensure the value is optimal. Also, few input features such as variance, minimal (full spine flexion angle), VAS, ODI, TSK PCS, PSEQ and CSQ do not show as optimal feature. This need to be clarified.

- Since the paper proposed using CNN, there is no information about the training of the CNN; how to avoid overfitting issue, how to ensure each of the training parameter of the CNN is optimal.

- Please explore whether the result is statistically significant.

Reviewer #2: This study aimed to develop and evaluate a machine learning model called "BACK-to-MOVE" for classifying non-specific low back pain (NSLBP) using standard video data, spinal motion data, and patient-reported outcome measures. The model utilized a Convolutional Neural Network architecture for effective pose estimation from video data. Results showed that the "BACK-to-MOVE" model successfully differentiated motor control impairment (MCI) and movement impairment (MI) classes with 93.98% accuracy, demonstrating strong criterion validity. The study concludes that automated classification of NSLBP using computer vision and machine learning could expedite personalized NSLBP rehabilitation management, potentially benefiting patients and healthcare services.

Experiment:

lack comparison among multiple CNNs performance on human pose detection (CrowPose, DensePose, PoseNet, UniPose, HigherHRNet), better provide a simple comparison.

Discussion:

Table 6, the model performance decreases when adding PROMs, this requires more in-depth discussion and ablation study to illustrate that whether this is caused by model. That is whether design a good model the results will change.

Minor error:

Line 76: MDC classification -> MDC, since C means classification.

The label of x and y-axis in Fig. 3 are too vague.

6. PLOS authors have the option to publish the peer review history of their article (what does this mean?). If published, this will include your full peer review and any attached files.

Reviewer #1: No

Reviewer #2: No

---

## [Author Response · Author response to Decision Letter 0]

4 Apr 2024

To PLOS1 Editor:

1. We note that you have indicated that there are restrictions to data sharing for this study. PLOS only allows data to be available upon request if there are legal or ethical restrictions on sharing data publicly. If there are ethical or legal restrictions on sharing a de-identified data set, please explain them in detail (e.g., data contain potentially identifying or sensitive patient information, data are owned by a third-party organization, etc.) and who has imposed them (e.g., a Research Ethics Committee or Institutional Review Board, etc.). Please also provide contact information for a data access committee, ethics committee, or other institutional body to which data requests may be sent.

Our response: 

Data sharing: Ethical restrictions exist related to the nature of the data (videos) containing potentially identifying and sensitive information (e.g. tattoos, birth marks, sex, ethnicity, unique movement characteristics etc) with no possibility to de-identify, add noise, or block portions of the dataset and still preserve the usefulness of the dataset to replicate the study results. Unfortunately, our institution (Cardiff University) does not have an established point of contact to field external requests for access to sensitive data.

Reviewer #1: 

1. It is not clear how to validate or ensure data MI and MCI at the beginning before this data fed to feature extraction process. 

MI/MCI classification is described in methods (clinical classification process lines 147-158). In short, MI/MCI was performed using clinical judgement-based MDC method, which is the only available method and considered as the gold standard. In our study, two MDC-trained expert clinicians independently classified the participants into MI and MCI following a process shown to have above 90% inter-examiner agreement [reference 15 and 27 in reference list]. We added additional detail to clarify the independent confirmation by the 2 experts (line 157). 

2. The structure of the CNN is not very clear such is the dimension of the input and output.

The structure of the HigherHRNet CNN, along with a detailed comparative analysis of HigherHRNet performance with that of alternative CNNs for human pose estimation and an ablation study, has been described previously and is referenced in our study [reference 47]. As the architecture of HigherHRNet is not the subject of this study, we limit ourselves to citing this work in our article. Nonetheless, to ensure clarity we added a sentence (line 199-201) referring readers to this detail.

3. Table 5 shows set of optimal input feature yielding the highest classification performance. Instead of using 'X' the actual metric such as classification should be used Moreover, how to ensure the value is optimal. Also, few input features such as variance, minimal (full spine flexion angle), VAS, ODI, TSK PCS, PSEQ and CSQ do not show as optimal feature. This need to be clarified. 

The features were optimal in the sense of producing the highest F1 measure value among all possible permutations of features, so using a numerical value against a single feature would not be possible as it is a metric for a feature subset. We added a sentence in the article (line 345-6) and a legend to Table 5 (line 387) to clarify the meaning. 

4. Since the paper proposed using CNN, there is no information about the training of the CNN; how to avoid overfitting issue, how to ensure each of the training parameter of the CNN is optimal. 

As already detailed in the manuscript (line 220-24) we utilised a CNN model (the selection of which is covered in our response to your comment 2 above) that was pre-trained on the MS-COCO dataset with fine-tuning not pursued given its superior performance and to prevent excessive overfitting. 

5. Please explore whether the result is statistically significant. 

Thank you for your comment. The aim of this study was to test the performance of the proposed classifier against a clinical judgement classification using MDC method as the ‘gold standard’. The conventional method for this form of performance evaluation is a calculation of accuracy, sensitivity and specificity, and F1-measure (Taha et al 2015, referenced in manuscript). There is no other model against which our proposed model could be benchmarked using statistical testing. However, to respond to your question, we explored the statistical significance of our results following a method outlined by Combrisson et al 2015 [reference 57 added], which showed that our results would reach statistical significance (p-value <0.001) for accuracy of 87.5% and higher. We added this detail and a reference in the article (added lines in methods 350-364, and results 405-407).

Reviewer #2:

1. Experiment: lack comparison among multiple CNNs performance on human pose detection (CrowPose, DensePose, PoseNet, UniPose, HigherHRNet), better provide a simple comparison. 

Thank you for your comment. A detailed comparative analysis of the performance of HigherHRNet with that of alternative CNNs for human pose estimation and an ablation study is described in detail in reference [47]. As CNNs performance comparison was not the aim of this study, we limit ourselves to referencing it [47] in our article. Nonetheless, to ensure clarity we added a sentence (line 199-201) referring readers to this detail.

2. Discussion: Table 6, the model performance decreases when adding PROMs, this requires more in-depth discussion and ablation study to illustrate that whether this is caused by model. That is whether design a good model the results will change. 

Thank you. In response to your comment, further text has been added to elaborate on the possible reasons for the limited discriminate ability of PROMs (line 463-70). In summary, PROMs in this study measure levels of pain intensity, disability, and psychological distress that vary according to different levels of pain intensity and duration. As participants with LBP > 3 months with pain intensity not precluding them from performing the movements were included, PROMs scores were unlikely to vary across the subgroups. Nevertheless, given the multi-factorial nature of LBP, we felt it was still important to explore any such variation as highlighted in the discussion section.

3. Minor error:

Line 76: MDC classification -> MDC, since C means classification. 

Thank you, ‘classification’ has now been removed. 

The label of x and y-axis in Fig. 3 are too vague. 

Figure 3 has been re-generated with Y axis titled as ‘Base of the neck vertical coordinate (pixels) and X axis titled ‘Base of the neck antero-posterior coordinate (pixels) for improved clarity.

---

## [Editor Report · Decision Letter 1]

16 Apr 2024

BACK-to-MOVE: Machine Learning and Computer Vision Model Automating Clinical Classification of Non-Specific Low Back Pain for Personalised Management.

PONE-D-23-28397R1

Dear Dr. Sheeran,

We’re pleased to inform you that your manuscript has been judged scientifically suitable for publication and will be formally accepted for publication once it meets all outstanding technical requirements.

Kind regards,

Adrian Pranata, Ph.D

Academic Editor

PLOS ONE

Additional Editor Comments (optional):

Dear Liba,

Thank you for addressing the comments from the reviewers and your patience. Research using AI in low back pain clinical management is scant. This study will add to the emerging area of AI for movement assessment and ultimately, management of people with low back pain. Thank you for your contribution in this important area of research.

Kind regards,

Adrian
---

## [Editor Report · Acceptance letter]

29 Apr 2024

PONE-D-23-28397R1 

PLOS ONE

Dear Dr. Sheeran, 

I'm pleased to inform you that your manuscript has been deemed suitable for publication in PLOS ONE. Congratulations! Your manuscript is now being handed over to our production team.

Kind regards, 

on behalf of

Dr. Adrian Pranata 

Academic Editor

PLOS ONE